# Effects of Vibrotactile Biofeedback Providing Real-Time Pressure Information on Static Balance Ability and Weight Distribution Symmetry Index in Patients with Chronic Stroke

**DOI:** 10.3390/brainsci12030358

**Published:** 2022-03-07

**Authors:** Ho Kim, Hongjun Kim, Won-Seob Shin

**Affiliations:** 1Department of Physical Therapy, Graduate School of Health and Medicine, Daejeon University, Daejeon 34520, Korea; hottirotti@naver.com; 2Department of Computer Engineering, Daejeon University, Daejeon 34520, Korea; hjkim99@dju.kr; 3Department of Physical Therapy, College of Health and Medical Science, Daejeon University, Daejeon 34520, Korea

**Keywords:** stroke, biofeedback, balance, sensor, rehabilitation

## Abstract

Training with visual and auditory biofeedback, in patients with stroke, improved balance ability and asymmetric posture. We developed a new biofeedback training device to prevent falls and improve balance ability in patients with stroke. This device corrects motion errors by collecting the pressure information of patients in real-time. This randomized crossover study aimed to investigate the effect of this biofeedback training on the static balance ability and weight distribution symmetry index in 24 patients with chronic stroke. Pressure sensor-based vibrotactile biofeedback, visual biofeedback providing posture information, and standing without biofeedback were randomly applied for 1 d each with 24 h washout intervals to minimize adaptation. The static balance ability was measured for each biofeedback training type, and the weight distribution symmetry index was calculated using the collected weight-bearing rate data. The static balance ability and weight distribution symmetry index differed significantly according to the type of biofeedback training used. Post-hoc analysis revealed significant differences in the order of newly developed vibrotactile biofeedback, visual biofeedback, and standing without biofeedback. These findings provide evidence that pressure sensor-based vibrotactile biofeedback improves static balance ability and weight support rates by proposing better intervention for patients with chronic stroke in the clinical environment.

## 1. Introduction

A stroke is a neurological dysfunction caused by a cerebrovascular accident or injury and is accompanied by sensory, motor, perception, and cognitive dysfunction [1]. It causes difficulties in balance and postural control, while impairments in standing and gait are caused by abnormal body balance, decreased ability to shift weight, and loss of motor control, which disturbs the performance of intricate functions [2,3]. Gait dysfunction is strongly related to balance disabilities [4].

Balance involves the planning and execution process in motor learning with the purpose of maintaining standing and sitting postures and maintaining posture homeostasis through a complex process of receiving and responding to sensory information [5]. To control balance ability, integration of the vestibular, visual sense, and somatosensory systems, and their interaction with the motor control system, musculoskeletal system, and cognitive ability are required [6]. In addition, balance ability involves the ability to control the center of gravity of the human body based on support in a given environment and plays a key role in performing all movements in daily life by keeping the body in a standing state [7]. However, in patients with stroke, postural sway increases due to impaired proprioceptive sense, and standing ability decreases due to asymmetric weight distribution and loss of weight-shifting ability. This has a profound influence on the person’s independence and gait ability in daily life. Therefore, proprioceptive sense, body stability, and symmetry are important treatment goals for patients with stroke [8]. Balance and enhancement of functional movement are important variables for rehabilitation in patients with stroke [9]. Providing additional sensations via the vestibular, visual, and somatosensory systems as biofeedback during the process of training balance and weight shifting in post-stroke patients is more effective in enhancing balance ability by activating sensation and motor integration in the central nervous system [10,11]. This helps to teach accurate movements by correcting errors while performing tasks by receiving feedback stimulation from the exercise performed by oneself or information related to movement from the vestibular, visual, and somatosensory systems [12].

Training with visual and auditory biofeedback in patients with stroke improved balance ability as well as asymmetric posture [13]. Exercise performed using a task-oriented approach focusing on biofeedback significantly improved the weight-bearing rate and balance ability toward the paralyzed side in a previous study [14]. Tactile biofeedback provides improved performance over cross-modality visual biofeedback or simultaneous visual and tactile feedback [15]. A 3-week somatosensory stimulation intervention given within 1 month after brain damage was effective in improving consciousness level and cognitive function in patients with acute brain damage [16]. However, there are insufficient studies that investigated the immediate effect of improving balance ability in patients with chronic stroke or by providing pressure information in real-time with feedback [17].

In this study, we provided real-time tactile biofeedback via pressure information given using a newly developed pressure sensor-based vibrating tactile biofeedback tool that directly corrects motion errors while a task if performed. We compared this with other biofeedback approaches, to identify the more effective method, by comparing the static balance ability and the immediate effect on the weight-bearing rate in patients with chronic stroke.

## 2. Materials and Methods

### 2.1. Participants

Thirty-nine patients with chronic hemiplegia among patients who were diagnosed with stroke admitted to Y Hospital in Daejeon, Republic of Korea, were recruited. The participant selection criteria were as follows: (1) patients with chronic stroke for >6 months after being diagnosed during stroke, (2) those who could take on an independent standing position for >5 min, (3) those with scores of <40 on the Berg Balance Scale (BBS), (4) those with scores of >24 on the Mini-Mental State Examination (MMSE), and (5) those who feel a sense of thickness <3.61 mm on the Semmes–Weinstein monofilaments test. Participants meeting any of the following criteria were excluded: (1) those with orthopedic problems that may affect the research, (2) those with visual impairments and deficiencies that could affect the research, and (3) those with communication problems [17]. Of the 39 recruited patients, 15 patients did not meet the selection criteria and were excluded. Finally, 24 patients with chronic stroke were selected for the analysis. 

The purpose and procedure of the study were explained to all participants before the study commenced, and written informed consent was obtained. All experimental procedures in this study were approved by our Institutional Review Board and are registered in the WHO International Clinical Trials Registry Platform: KCT0005399.

### 2.2. Study Design and Process

In this randomized crossover study, the order of measurement was randomized by inputting equations through the R Studio Desktop 1.2.5033 (R Studio, Inc., Boston, MA, USA) software for the 24 selected participants. We used pressure sensor-based vibrotactile biofeedback, visual biofeedback providing posture information, and a standing position without biofeedback randomly for 1 day each, with a 24-h washout period in-between, to avoid adaptation [18]. We aimed to determine whether there was a difference in the static balance ability and the weight distribution symmetry index of the standing posture after the use of each of these biofeedback conditions. 

Static balance ability was measured by maintaining a standing position for a total of 3 sets of 30 s with both feet naturally positioned on a Wii balance board (Nintendo, Kyoto, Japan), with 3 min rest time between sets. The distance from the Wii balance board to the wall was 2 m. The experiment was conducted barefoot. In all cases, the foot position was placed in the center of the left and right compartments on the Wii balance board (Figure 1).

### 2.3. Various Biofeedback Conditions

#### 2.3.1. Pressure Sensor-Based Vibrotactile Biofeedback

For this study, we developed a pressure sensor-based vibrotactile biofeedback system that consisted of the following substructures: (1) Arduino-UNO microcontroller (Arduino UNO R3; https://www.arduino.cc, accessed on 8 February 2022), (2) 4 pieces of vibrotactile motors (10 × 2.7 mm Coin Mobile Phone Vibration Motor), and (3) 4 piece of Flexiforce pressure sensors (Tekscan Inc., Boston, MA, USA). To measure the participants’ pressure information in real-time, four pressure sensors were placed on each foot below the great toe, the 1st and the 5th head of the metatarsal bone, and the bottom of the heel. To provide tactile biofeedback from this pressure information, vibration motors were placed at the midway point of the center of the anterior, lateral, medial, and posterior parts of the calves using Velcro [19]. When the participant’s torso was tilted to one side, the equipment attached to the leg in the same direction started to vibrate. Therefore, the participants aimed to maintain a straight posture so that the vibration did not produce any sound, and then stood on the Wii balance board with both feet, facing forward, after which their balance was measured (Figure 2A).

#### 2.3.2. Visual Biofeedback Providing Posture Information

Visual biofeedback is a rehabilitation treatment used for restoring dysfunction in patients with chronic stroke, providing active movement, improving muscle strength, and enhancing balance [20,21]. Visual biofeedback is an effective method to make the standing posture symmetrical, and real-time application has yielded marked improvement in physical ability [22,23].

In this study, visual biofeedback was measured by the participant standing on a Wii balance board while receiving visual feedback through a full-length mirror that was placed 2 m in front of the participant. The participants balanced while attempting to maintain and correct the correct posture through a mirror in which the whole body was visible in an upright posture and stood on the Wii balance board with both feet, looking forward. Thereafter, their balance was assessed (Figure 2B).

#### 2.3.3. Normal Standing Position with No Biofeedback

Without biofeedback, the balance was measured while standing upright on the Wii balance board by looking at a white wall, which was located 2 m from the participant, similar to the other group (Figure 2C).

### 2.4. Outcome Measures

#### 2.4.1. Static Balance Ability Assessment

The Wii balance board used to evaluate static balance is a device that allows load cells located at four corners to collect center of pressure (COP) data continuously and can be connected to the device through Bluetooth communication. The sampling rate of the collected data was controlled by connected software. The intra-measurement reliability, based on measurement and re-measurement of results obtained via the Wii balance board yielded an intraclass correlation coefficient (ICC) range of 0.79–0.93, and the ICC of the reliability between the measurements was in the range 0.79–0.95, indicating a very high level of confidence [23]. Balancia software (version 2.0; Mintosys, Seoul, Korea) was used to analyze the COP information of the static balance. Analysis of the COP information measured by the Wii balance board shows the moving distance and speed for the X and Y axes of the COP. All data were sampled and extracted at 100 Hz and filtered using a 10 Hz low-pass filter. The validity of this program was 0.85–0.96 and the reliability was 0.79–0.96, indicating that it was highly reliable for evaluating balance ability (Figure 3) [24].

#### 2.4.2. Weight-Distribution Symmetry Index

The weight support rate data extracted from the Wii balance board and Balancia software were used as measurement tools to analyze the left–right weight distribution symmetry index for the standing posture. To determine the symmetry index between the paralyzed side and the non-paralyzed side, the calculation formula of SI = (non-paralytic side-paralyzed side)/(non-paralytic side + paralyzed side) × 2 × 100 was used. The range of the symmetry index was from −200% to 200%, and 0% means that the left and right weight-support rates are the same. A negative value indicated that the paralytic side weight-support rate was high, indicating that the weight was further shifted to the paralyzed side [25].

### 2.5. Data Analysis

Data were analyzed using IBM SPSS software version 25.0 (IBM Co., Armonk, NY, USA). For the general characteristics of the participants, the mean and standard deviation were presented using descriptive statistics, and a normality test was performed using the Shapiro–Wilk test. One-way ANOVA with repeated measures was performed to determine the effect of each biofeedback approach and the weight-distribution symmetry index. Bonferroni correction was used for post hoc analysis. The statistical significance level was set at α = 0.05.

## 3. Results

### 3.1. General Characteristics of the Participants

The general characteristics of the participants including sex, paretic side, age, height, weight, duration of onset, MMSE score, and BBS score are presented in Table 1.

### 3.2. Difference in Static Balance Ability According to Various Biofeedback Conditions

#### 3.2.1. Sway Length

Significant differences in the sway length were observed according to various biofeedback conditions (F (56.857), *p* < 0.001). The average sway length related to the tactile biofeedback was 91.11 ± 19.27 cm, which was the smallest movement distance among the various biofeedback conditions. Post-hoc analysis revealed significant differences in sway length in the order of tactile biofeedback, visual biofeedback, and no biofeedback (*p* < 0.001) (Table 2).

#### 3.2.2. Sway Velocity

Significant differences in the sway velocity were observed according to various biofeedback conditions (F (38.382), *p* < 0.001). The average sway velocity related to the tactile biofeedback was 3.06 ± 0.87 cm/s, which was among the biofeedback conditions. Post-hoc analysis revealed significant differences in sway velocity in the order of tactile biofeedback, visual biofeedback, and no biofeedback (*p* < 0.001) (Table 2).

#### 3.2.3. Weight-Distribution Symmetry Index

Significant differences were found in the weight distribution symmetry index among the various biofeedback conditions (F (41.663), *p* < 0.001). The average of the weight distribution symmetry index of tactile biofeedback was 10.85 ± 9.81. Thus, the distribution of the weight-bearing ratio was the most constant for this condition among the various biofeedback conditions. Post-hoc analysis revealed significant differences in weight-support in the order of tactile biofeedback, visual biofeedback, and no biofeedback (*p* < 0.001) (Table 3).

## 4. Discussion

The purpose of this experimental study was to compare different biofeedback conditions and identify which biofeedback approach had a greater immediate effect on the static balance ability and weight distribution symmetry index in 24 patients with chronic stroke.

When we measured the static balance ability, the pressure sensor-based vibrotactile biofeedback in the sway length was significantly less than that of conventional visual biofeedback and no biofeedback (*p* < 0.001). Additionally, pressure sensor-based tactile biofeedback had a significantly slower sway velocity than the existing visual biofeedback and no biofeedback (*p* < 0.001). For static balance ability, the longer the sway length and the faster the sway velocity, the lower the static balance ability was [24]. Therefore, these results show that the pressure sensor-based tactile biofeedback approach proposed in this study can be more effective than other biofeedback approaches and can further promote the static balance ability of patients with chronic stroke. We believe that it is possible to correct balance errors rapidly through immediate vibration stimulation by facilitating the recognition of changes in the center of gravity in patients with chronic stroke in real-time by receiving pressure information. Therefore, the balance could be controlled while maintaining a more accurate posture compared with that under other biofeedback conditions for patients with stroke. Balance training using biofeedback is effective in improving balance ability and enhances lower extremity muscle activity of patients with stroke better than does general treatment [26,27]. Among the different treatments, vibrotactile biofeedback can prevent falls by providing augmentative sensory information on the direction and intensity of imbalance and result in the immediate improvement of task performance as compared to providing visual biofeedback. It could be an effective tool for improving motor relearning in patients with stroke [28,29]. In addition, when electrical tactile stimulation is applied to the peroneal nerve in patients with stroke, spasticity is reduced due to reinforcement of presynaptic inhibition of the spastic plantar flexor. Repeated tactile stimulation improves neuromotor function based on multisensory integration, thereby improving postural instability [30]. This tactile biofeedback activates various afferent nerve fibers, including the proprioceptive sense, increases the influx of somatic sensation of the lower extremity, reorganizes the motor function area of the brain, and increases motor output, which is thought to affect static balance ability [31,32].

When we measured the weight-distribution symmetry index in this study, pressure sensor-based vibrotactile biofeedback showed a significant difference from the effect of visual biofeedback and no biofeedback (*p* < 0.001). This showed the same improvement in body stability as in previous studies that provided tactile biofeedback through vibrational sensory signals [33]. Tactile biofeedback is reported to increase the quality of gait patterns, improve symmetry index, and promote motor learning [34]. The uniformity of the lower extremity symmetry increases the stability of the lower extremity and has a positive effect on balance ability [35]. Several clinical studies have confirmed that the cerebral cortex is reorganized through stimulation of the peripheral nerves, through an environment presenting considerable stimulation, and through the active use of the lesion side [36]. The mechanism for improving sensory stimulation-based mediation involves neuronal plasticity in the central nervous system. In other words, the neurons damaged by the stroke are not regenerated, but the external stimulus introduced into the cerebral cortex through sensory stimulation activates the sensory receptors, which changes the sensory acceptance ability of the brain and leads to the reconstruction of the cerebral cortex [37,38]. It was observed that when stimulation was applied to the peripheral part, several parts of the cerebral cortex were activated [36]. As such, tactile biofeedback training applied to patients with stroke can be used as a valid tool for physical therapists [39]. Recently, vibrotactile biofeedback has been used to improve gait symmetry training in patients with stroke with hemiplegia, which supported our findings [40]. Therefore, we expect that the pressure sensor-based vibrotactile biofeedback will have a positive effect on the improvement of the weight support rate of patients with chronic stroke as compared to other biofeedback conditions that are implemented in clinical practice. Visual biofeedback stimulates the paralyzed side of a patient with stroke and induces weight bearing on the paralyzed side [41]. In addition, it yielded better functional improvement than no biofeedback therapy by promoting motor learning [42,43]. In a previous pilot study, the vibrotactile biofeedback device was compared with that of a control group in which the intervention was not applied [17]. However, in the present study, we compared it with the visual biofeedback intervention using a mirror, which is actually applied in clinical practice. Therefore, the vibrotactile biofeedback device developed by us is expected to provide a more useful direction in clinical practice.

There were some limitations to this study. First, the age of patients with chronic stroke was uneven. Second, because of the characteristics of the pressure sensor-based vibrotactile biofeedback tool, the dynamic balance ability was not evaluated; therefore, the effect on the dynamic balance ability is unclear. Third, since this study was conducted to compare immediate effects, future studies will need to evaluate the effects of the relevant mid-to-long-term interventions.

## 5. Conclusions

This experimental study was conducted to investigate the immediate effect of the newly developed pressure sensor-based vibrotactile biofeedback tool on 24 patients with chronic stroke by comparing the static balance ability and weight distribution symmetry index of this approach to those of other biofeedback conditions commonly used in clinical practice. Our results confirm that the pressure sensor-based vibrotactile biofeedback had a more significant impact on the improvement of static balance and weight-distribution symmetry index than did other biofeedback conditions. These findings provide evidence that pressure sensor-based vibrotactile biofeedback improves static balance ability and weight support rates by proposing better intervention for patients with chronic stroke in the clinical environment.

## Figures and Tables

**Figure 1 brainsci-12-00358-f001:**
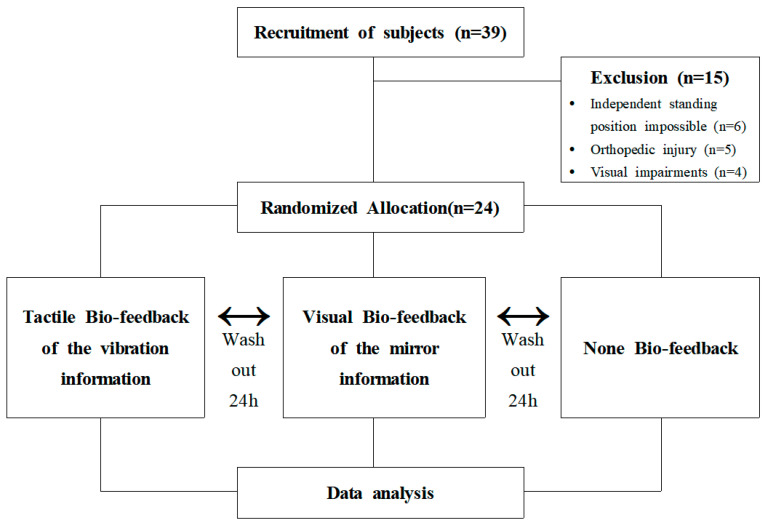
Flow chart.

**Figure 2 brainsci-12-00358-f002:**
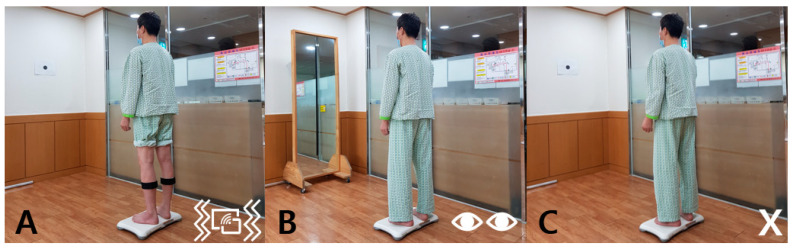
Various bio-feedback conditions (**A**) Pressure sensor-based vibrotactile biofeedback, (**B**) Visual biofeedback providing posture information, (**C**) Normal standing position with no biofeedback).

**Figure 3 brainsci-12-00358-f003:**
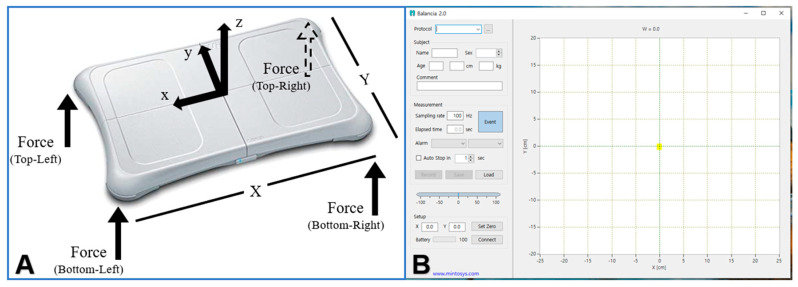
Static balance ability and weight bearing rate measurement tool ((**A**) Wii balance board, (**B**) Balancia software ver 2.0).

**Table 1 brainsci-12-00358-t001:** The general characteristics of subjects.

Variables	Values
Sex (Male/Female)	18/6
Paretic side (left/right)	10/14
Age (year)	63.00 (12.31)
Height (cm)	163.08 (8.84)
Weight (kg)	59.25 (8.83)
Duration of onset (month)	15.54 (9.00)
MMSE (score)	26.54 (2.34)
BBS (score)	34.38 (4.23)

Values are expressed as Mean (Standard Deviation).

**Table 2 brainsci-12-00358-t002:** Comparison of static balance according to the various bio-feedbacks.

Variables	Tactile Biofeedback	Visual Biofeedback	No Biofeedback	F
Sway Length (cm)	91.11 (19.27) ^†‡^	102.89 (28.66) ^‡^	114.59 (28.78)	56.857 *
Sway Velocity (cm/s)	3.06 (0.87) ^†‡^	3.34 (1.10) ^‡^	3.68 (1.09)	38.382 *

Values are expressed as Mean (Standard Deviation). * *p* < 0.001. ^†^ Significant difference (*p* < 0.001) from Visual Biofeedback. ^‡^ Significant difference (*p* < 0.001) from No Biofeedback.

**Table 3 brainsci-12-00358-t003:** Comparison of weight distribution symmetric index according to the various bio-feedbacks.

Variables	Tactile Biofeedback	Visual Biofeedback	No Biofeedback	F
Weight-distribution symmetric index (%)	10.85 (9.81) ^†‡^	25.89 (17.65) ^‡^	39.21 (24.42)	41.663 *

Values are expressed as Mean (Standard Deviation). * *p* < 0.001. ^†^ Significant difference (*p* < 0.001) from Visual Biofeedback. ^‡^ Significant difference (*p* < 0.001) from No Biofeedback.

## Data Availability

The data presented in this study are available on request from the corresponding author. The data are not publicly available due to data privacy regulations.

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
