# Peer review of "Effects of Vibrotactile Biofeedback Providing Real-Time Pressure Information on Static Balance Ability and Weight Distribution Symmetry Index in Patients with Chronic Stroke"

_brainsci, 2022, doi:10.3390/brainsci12030358_

Round 1
Reviewer 1 Report
Summary: Patients afflicted with stroke often suffer from long-term motor dysfunction including their ability to control gait and balance. Rehabilitation of motor function often relies on vestibular training as well as visual, auditory, and tactile biofeedback. This study uses a real time vibrating tactile biofeedback device to correct errors in motion encountered while performing a specific task. The purpose of this study was to compare the effectiveness of various biofeedback modalities on static balance ability and weight distribution symmetry in 24 patients with chronic stroke. The results show that sway length and sway velocity was lowest in vibrotactile biofeedback, followed by visual biofeedback and no biofeedback. Similarly, weight distribution was most symmetrical with vibrotactile biofeedback, followed by visual biofeedback and no biofeedback. Therefore, the study demonstrates that the pressure sensor-based vibrotactile feedback tool helps improve static balance ability and weight bearing in the short-term and may be used as a clinical intervention for patients with chronic stroke.
Strengths: One of the strengths of this study is that it addresses an important patient population – those with chronic disability following stroke. The randomized crossover design of the study also adds to its strength. The study design eliminates the effect of patient variation on the interventions under consideration.
Limitations: The study has a relatively small sample size. Authors also acknowledge other limitations including inability to measure dynamic balance and evaluate long-term effects of the vibrotactile intervention.
Minor Comments:
- Please provide some information about the vibrotactile feedback device used in this study (either in the Introduction or Discussion sections). Who is the manufacturer? Were other pre-clinical or clinical studies performed using this same device and what were the outcomes? Or is this a usability study testing the device prototype for the first time in patients?
- Page 6, Line 185: Incorrectly states average sway “length”. It should be average sway “velocity”.
- Page 7, Line 211-216: Description of the Semmes-Weinstein monofilament test seems out of place in the Discussion. Please consider moving it to the Methods section under 1. Participants. Also consider adding similar rationales for the other evaluations used for participant inclusion/exclusion, i.e., BBS and MMSE.
Reviewer 2 Report
Moderate English changes required
Reviewer 3 Report
There are many instances of English grammar errors. I recommend having someone in which English is their first language read and edit the manuscript.
Abstract-States the “we developed a widely used biofeedback device” but a commercial Wii balance board was used in the study. Line 66 states a “newly developed pressure sensor-based vibrating tactile biofeedback tool (again this refers to a commercially available Wii board)
Line 34 “Gait damage” should be “Gait dysfunction”
Line 64 References should be included.
Line 96 Please state whether shoes and socks were removed.
Line 98 It is unclear what “all participants were politely positioned” means.
Line 122 “Participants measured balance” needs rewording as participants were not measuring anything.
Sampling rate of data collection was not described. Need to contact the manufacturer as this information is important to include.
Lines 210-215 should not be in the Discussion but rather Methods section.
Lines 253 It is unclear what is meant by the “cerebral cortex was reconstituted through stimulation”
Lines 273-Rephase that “the age of the patients with chronic stroke was uneven”
These results are very transient in nature, thus the statements that pressure sensor-based vibrotactile biofeedback will help improve static balance ability and weight support rates and help improve the quality of life” are not substantiated.
Round 2
Reviewer 3 Report
Revisions are adequate and improved the manuscript.